# Calcium Enhances Thallium Uptake in Green Cabbage (*Brassica oleracea* var. *capitata* L.)

**DOI:** 10.3390/ijerph20010004

**Published:** 2022-12-20

**Authors:** Yanlong Jia, Tangfu Xiao, Jialong Sun, Zengping Ning, Enzong Xiao, Xiaolong Lan, Yuxiao Chen

**Affiliations:** 1School of Chemistry and Environmental Engineering, Hanshan Normal University, Chaozhou 521041, China; 2School of Resources and Environmental Engineering, Guizhou Institute of Technology, Guiyang 550002, China; 3School of Environmental Science and Engineering, Guangzhou University, Guangzhou 510006, China; 4State Key Laboratory of Environmental Geochemistry, Institute of Geochemistry, Chinese Academy of Sciences, Guiyang 550002, China

**Keywords:** thallium-contaminated soil, calcium, green cabbage, phytoextraction, sequential extraction

## Abstract

Thallium (Tl) is a nonessential and toxic trace metal that is detrimental to plants, but it can be highly up-taken in green cabbage (*Brassica oleracea* L. *var. capitata* L.). It has been proven that there is a significant positive correlation between Tl and Calcium (Ca) contents in plants. However, whether Ca presents a similar role for alleviating Tl toxicity in plants remains unclear, and little is known in terms of evidence for both Ca-enhanced uptake of Tl from soils to green cabbage and associated geochemical processes. In this study, we investigated the influence of Ca in soils on Tl uptake in green cabbage and the associated geochemical process. The pot experiments were conducted in 12 mg/kg Tl(I) and 8 mg/kg Tl(III) treatments with various Ca dosages. The results showed that Ca in soils could significantly enhance Tl uptake in green cabbage, increasing 210% in content over the control group. The soluble concentrations of Tl were largely increased by 210% and 150%, respectively, in 3.0 g/kg Ca treatment, compared with the corresponding treatment without Ca addition. This was attributed to the geochemical process in which the enhanced soluble Ca probably replaces Tl held on the soil particles, releasing more soluble Tl into the soil solution. More interestingly, the bioconcentration factor of the leaves and whole plant for the 2.0, 2.5, 3.0 g/kg Ca dosage group were greatly higher than for the non-Ca treatment, which could reach 207%, implying the addition of Ca can improve the ability of green cabbage to transfer Tl from the stems to the leaves. Furthermore, the pH values dropped with the increasing Ca concentration treatment, and the lower pH in soils also increased Tl mobilization, which resulted in Tl accumulation in green cabbage. Therefore, this work not only informs the improvement of agricultural safety management practices for the farming of crops in Tl-polluted and high-Ca-content areas, but also provides technical support for the exploitation of Ca-assisted phytoextraction technology.

## 1. Introduction

As a nonessential toxic heavy metal that is harmful to plants [1,2], thallium (Tl) is not only more toxic than lead (Pb) and mercury (Hg), but also easily accumulated in the human body [3,4]. The minimum lethal dose of Tl (salt) in adults ranges from 0.1 to 0.7 g, usually from 0.5 to 1 g [5]. The lethal dose 50 of Tl in rats ranges from 30 to 32 mg/kg, lower than Hg (39.6 mg/kg) and Pb (4665 mg/kg) [6,7]. Tl is categorized as one of thirteen metal pollutants of importance in America [8] and a regulated metal pollutant in China [9,10]. While thallium is naturally widely occurring, average Tl soil concentrations globally are less than 1 mg/kg [11]. A guideline for the environmental quality of Tl in soil is thus far only available in Canada [12]. However, elevated concentrations of Tl have been detected in the soil in various regions [13,14,15,16,17,18,19,20,21,22,23,24,25,26,27,28,29,30,31,32], due to either pedogeochemical or anthropogenic (i.e., mining, processing, and smelting) sources.

According to earlier reports, particular plants exhibit increased susceptibility to the accretion of Tl from Tl-polluted soils [33,34,35,36,37,38]. Among them, green cabbage (*Brassica oleracea* L. *var. capitata* L.), a common vegetable crop, has been identified as a crop plant prone to highly uptake Tl from soils [15,36,39,40]. Calcium (Ca) is an essential macronutrient for both plants and animals. It was, interestingly, observed that the uptake of Tl had a significantly positive correlation with Ca contents in both whole leaf tissue [15] and subcellular cytosol and vacuole fraction in the leaves of green cabbage [41]. Similarly, a positive correlation was found between Tl and Ca in the main vein of the basal leaves of Tl-hyperaccumulator *Iberis intermedia* [42,43]. Some previous studies have demonstrated that Ca can assist in the alleviation of Cd, Pb, As, and Cu toxicity, improving plant growth in contaminated soils [44,45,46,47]. All these studies suggested that Ca may play an important role in enhancing Tl uptake in green cabbage.

To date, however, whether Ca presents a similar role for alleviating Tl toxicity in plants remains unclear, and little is known in terms of evidence for both Ca-enhanced uptake of Tl from soils to green cabbage and associated geochemical processes. The objective of this study was to characterize the influence of Ca in soils on Tl uptake in green cabbage through pot-culture trials, and to gain new insight into the geochemical processes of Ca for constraining Tl uptake to plants. The findings can inform the improvement of agricultural management practices for the farming of crops in Tl-polluted areas and phytoremediation.

## 2. Materials and Methods

### 2.1. Pot Trials

Soils used in pot experiments were obtained from the upper layer (0–20 cm) of a slope in an area unpolluted with Tl in the suburbs of Guiyang City (106°42′ E, 26°34′ N), Guizhou Province, Southwest China. The physicochemical properties of the pot soils are summarized in Table 1. The initial soil collected for the experiment was classified as weak acidic (pH values 6.27) and characterized by low CEC (mean at 21.5 cmol/kg) and high SOM (mean at 70.3 g/kg). The exchangeable Ca concentration in the initial soil was 660 mg/kg, averaging 6.70% of the total Ca. The total Tl concentration in the initial soil was 0.85 mg/kg, which was within the background value of Chinese soils (0.29–1.17 mg/kg Tl) [48].

The pot soil was mainly composed of quartz and smectite, and the detailed mineral compositions are summarized in Table 2. Stones, plant roots, and other sizable debris were removed from the soil by passing through a 2 mm stainless-steel sieve, following which the soil was air-dried and combined.

Two oxidation states, namely monovalent Tl(I) and trivalent Tl(III), exist in Tl and differ with respect to their toxicity and chemical reactivity [49]. Although Tl(III) may be stabilized by hydrolysis and colloid formation or sorption to Fe(III)-colloids, Tl(I) tends to dominate over Tl(III), due to the high redox potential of Tl(III)/Tl(I) couple (Eh = 1.28 V) [50,51]. Thus, Tl was artificially combined with each soil sample as Tl^+^ (12 mg/kg, TlCl) and Tl^3+^ (8 mg/kg, Tl(NO_3_)_3_·3H_2_O). A control treatment without Tl spike was carried through the experiment. Following three saturation cycles (15 d per cycle) with deionized water (aiding the Tl fraction distribution to obtain a relative balance) and 2.5 kg air-drying, soil samples were added to plastic pots (25 cm height, 20 cm diameter). Prior to seeding, 10 g of soil was collected from each pot for Tl fractionation and pH assessment. Green cabbage seeds were obtained from the Vegetable Research Institute of Guizhou Agricultural Academy, China. Once the seeds had been superficially sterilized in 0.5% sodium hypochlorite (NaOCl) and thoroughly rinsed with deionized water, they were placed onto a feeding block contained in a climate chamber (RXZ-300C-type). After approximately 5 d, plants exhibiting similar growth were transferred into each pot (one plant in one pot). One week later, the soil was supplemented five times with Ca^2+^ (Ca(NO_3_)_2_·4H_2_O) to obtain Ca concentrations of 1.0, 1.5, 2.0, 2.5, and 3.0 g/kg. The treatments were performed in triplicate. The plants were cultivated for 12 weeks in a plant growth chamber (temperature of 25–30 °C, humidity of 40–60%). Tap water was added to the pots to ensure that the moisture level was just less than field water capacity to prevent leaching from the pots. Each pot was fertilized approximately once a week with 0.1 L 50% Hoagland’s solution [52] lacking Ca(NO_3_)_2_·4H_2_O and trace element solution for each pot.

Following the three-month growth phase, the green cabbage samples were gathered and divided into root, stem, and leaf samples, which were then rinsed with deionized water to eliminate Tl contamination from dust or soil particles on the plants. The samples were then air-dried at 80 °C for 72 h. Plant tissue dry weight was then calculated. The plant samples were then pulverized using a micro plant grinding apparatus (FZ102, TAISITE, Tianjin, China) and passed through a 60-mesh sieve. Selected rhizosphere soil samples were also obtained during plant tissue collection. A shaking-off method [53] was applied to obtain the rhizosphere soil samples. All of the soil samples were air-dried, powdered in a ceramic disc mill, and passed through a 100-mesh sieve.

### 2.2. The Exchangeable Ca in Ca-Spiked Soils

Ten grams of pot soil was weighed into 50 mL polypropylene centrifuge tubes. Ca^2+^ (Ca(NO_3_)_2_·4H_2_O) was added to the soils to obtain Ca concentrations at 0.5, 1.0, 1.5, 2.0, 2.5, 3.0 g/kg, respectively, while deionized water was used for the controls. After undergoing three cycles (15 d per cycle) of saturation with deionized water and air-drying, 2.5 g of the soil was weighed into 50 mL polypropylene centrifuge tubes, and 25 mL of a 1 mol/L ammonium acetate was added to the tubes. The tubes were then rotated in a shaker at 120 rpm for 1 h, and the supernatant was separated after passing through a 0.45 μm microfilter [54].

### 2.3. The Solubility of Tl in Ca Solution

Three levels of Ca^2+^ solutions (1.0, 2.0, and 3.0 g/kg) were added to the pot soils with 12 mg/kg Tl(I) and 8.0 mg/kg Tl(III), respectively, while deionized water was used for the controls. After undergoing three cycles (15 d per cycle) of saturation with deionized water and air-drying, 1 g of the soil sample was weighed into 50 mL polypropylene centrifuge tubes, and 25 mL H_2_O added. The tubes were then rotated in a shaker at 120 rpm for 1 h, and the supernatant was separated after passing through a 0.45 μm microfilter. Each treatment was replicated three times.

Tl-polluted soil sample, which contained Tl at 87.5 mg/kg from the Lanmuchang Tl-Hg-As mineralized area [55], was also weighed into 50 mL polypropylene centrifuge tubes. The 25 mL Ca^2+^ extractant solutions (1.0, 2.0, and 3.0 g/L, respectively) were added to the above soils, while deionized water was used for the controls. The following procedure was performed as mentioned above. Each treatment was replicated three times.

### 2.4. Sequential Extraction of Tl in Soils

The complete procedure for the sequential extraction of Tl in soil samples, i.e., the separation of Tl into water soluble, weak acid soluble, reducible, oxidizable, and residual fractions, was achieved according to a BCR procedure, as described by Rauret et al. [56] with little modification [57,58,59], which was also adopted to quantify the bioavailability of Tl. In the first step (water soluble), 1.0 g of the sieved soil sample (<74 μm) was extracted using 30 mL deionized water. The mixture was then stirred for 1 h at 22 ± 5 °C, centrifuged at 3000× *g* for 20 min, and then the supernatant was collected and filtered through syringe filter with 0.45 μm nitrocellulose membrane. In the second step (weak acid soluble), the residue from step 1 was shaken for 16 h at 22 ± 5 °C with 0.11 mol/L acetic acid, centrifuged (3000× *g*, 20 min), and decanted. The solid residue was washed with 20 mL of deionized water, shaken for 15 min, centrifuged (3000× *g*, 20 min), and the supernatant collected combined with the extract. The above washing step was repeated, and the residue was decanted and filtered as in step 1. In the third step (reducible), the residue from step 2 was shaken for 16 h at 22 ± 5 °C with 40 mL 0.5 mol/L hydroxylamine hydrochloride, acidified to pH 1.5 with HNO_3_ and centrifuged, decanted, and filtered as in step 1. The residue was washed with deionized water as in step 2. In the fourth step (oxidizable), 10 mL of 8.8 mol/L hydrogen peroxide, acid-stabilized with HNO_3_ to pH 2–3, was carefully added to the residue from step 3. The tube was covered with a watch glass and the contents digested at 22 ± 5 °C for 1 h with occasional manual shaking. The digestion was continued in a water bath for 1 h at 85 ± 5 °C and the volume was reduced to approximately near dryness by further heating of the uncovered tube. Then, a further 10 mL of hydrogen peroxide was added, and the tube was covered with a watch glass and again heated to 85 ± 5 °C. After 1 h, the cover was removed. To the cool moist residue, 50 mL of 1 mol/L ammonium acetate with HNO_3_ to pH 2 was added, shaken for 16 h at 22 ± 5 °C, centrifuged, decanted, and filtered as in steps 1 and 2. In the fifth step (residual), the residue from step 4 was digested in a heated acid mixture (15 mL of 15 M HNO_3_ and 5 mL of 10 M HF) to calculate total Tl. To assess the degree of experimental contamination, blank experiments were executed under identical settings. The results indicated that no noteworthy Tl contamination had occurred during the sequential extraction. Tl recovery was calculated by evaluating the quantity of Tl extracted against the total amount indicated by its total digestion concentration: (sum/total) × 100%. The recovery rate of Tl ranged from 102–112% in the present study.

### 2.5. Analysis and Quality Control

Soil pH was measured using a pH meter once the soil had been suspended in deionized water at a ratio of 1:2.5 mass/volume (AISI pHB9901, Taiwan, China). Soil organic material (SOM) was calculated by catalytic oxidation (1350 °C) using both the Metalyt CS 500 and Metalyt CS 530 elemental analyzers (Eltra, Hamburg, Germany). The cation exchange capacity (CEC) was determined after saturating the soil samples in a 0.005 M EDTA and 1 M ammonium acetate mixture, followed by titration with ammonia acid. The exchange extractant sodium acetate (1 mol/L) reacted with the Ca^2+^ in the soil, and the exchangeable Ca^2+^ in the solution was determined using inductively coupled plasma spectrometry (ICP-OES, iCAP 6500, Thermo Scientific, Karlsruhe, Germany) following filtration.

About 50 mg of the sieved soil sample (<180 µm) was digested in a heated acid mixture (15 mL of 15 M HNO_3_ and 5 mL of 10 M HF) to calculate total Tl and Ca. Furthermore, macerated plant sample (100 mg) was processed with 10 mL strong acid mixture (8 mL of 15 M HNO_3_ and 2 mL of 10 M HF) for total Tl determination. Tl was measured using ICP-MS (PerkinElmer, ELAN DRC-e, Durham, NC, USA) by standard addition with Rh (10 μg/L) as an internal standard [59].

The soil and plant sample detection limit of total Tl was 0.01 mg/kg. The analytical precision, which was assessed based on typical quality control practices that utilize internationally certified reference materials (NIST 2711 and GBW07405), duplicates, and reagent blanks, was less than ±10%. The plant data are reported as dry weight (DW).

### 2.6. Statistical Analysis

The impacts of the treatments on pH, extractable Tl, plant Tl, and plant biomass were assessed using SPSS 17.0 for Windows. Least significant difference (LSD) analysis was used to evaluate treatment differences at the 5% significance level. We employed a type I error (α) of 5% for all of the statistical evaluations.

## 3. Results and Discussion

### 3.1. Ca-Induced Plant-Enhanced Tl Accumulation

The biomass of the roots, stems, young leaves, old leaves, and whole plants of green cabbage from the pot trials is summarized in Table 3. Under 12 mg/kg Tl(I) or 8 mg/kg Tl(III) treatment, the dry weights of the roots, stems, young leaves, old leaves, and whole plants are significantly different between Ca-treated trials and controls, except for the dry weight of the roots at 8 mg/kg Tl(III) treatment. A significantly positive correlation between the dry weight of leaves and whole plants and the addition of Ca was observed (Figure 1), indicating that the addition of Ca leads to an increase for the biomass of leaves and whole green cabbage plants.

Green cabbage biomass was maximum at the Ca treatment of 2.0 g/kg. There was no significant difference among Ca treatments (*p* < 0.05), which may be attributed to the fact that the exchangeable Ca concentration under 2.0 g/kg Ca treatment was saturated in the soil (Figure 2).

The total Tl concentration in the roots, stems, and leaves of green cabbage in the pot experiments is summarized in Table 4. For 12 mg/kg Tl(I) and Ca-spiked treated pot trials, the total Tl concentration increased by a maximum of 31.7%, 26.2%, 104%, and 88.7% in the roots, stems, leaves, and whole plants of green cabbage for the Ca-treated pots, respectively, relative to the non-Ca-treated control. For 8 mg/kg Tl(III) and Ca-spiked treated pot trials, the total Tl concentration increased by a maximum of 24.1%, 7.33%, 60.5%, and 57.4% in the roots, stems, leaves, and whole plants of green cabbage for the Ca-treated pots, respectively, relative to the non-treated control. It is interesting to observe that the total Tl concentration was significantly higher in the roots of green cabbage from the non-Ca-treated control than in the 1.0 g/kg Ca treatment. Similar results were also observed in the stems of green cabbage in 1.0 g/kg and 1.5 g/kg Ca-treated pots.

The enrichment of Tl in the leaves and whole plants of green cabbage was positively correlated with Ca treatment in the pot soils (*p* < 0.05), which can be attributed to the fact that the leaves of green cabbage were the major storage sites (greater than 80%) for Tl [55]. On the contrary, no similar correlations were observed for the enrichment of Tl in the roots and stems of green cabbage (Figure 3).

The effect of Ca on the total Tl mass of the plants is shown in Figure 4. The total Tl mass of green cabbage was increased by a maximum of 210% and 160% for the Ca-treated pots with 12 mg/kg Tl(I) and 8 mg/kg Tl(III) treatment, respectively, relative to the non-Ca-treated control.

The transfer coefficient (TC) is the ratio of an element concentration in aboveground plant tissue in comparison to its root concentration. TC is used to evaluate the capacity of aboveground plant tissue to transport an element from the roots. The bioconcentration factor (BF) is the ratio of an element’s concentration in plant tissue to its concentration in rhizospheric soils and is used to estimate the ability of the plant to accumulate the element. The TCs and BFs for green cabbage in the pot experiments are summarized in Table 5. Higher TCs and BFs were observed in the leaves. In addition, the BFs of leaves and whole plant at 8 mg/kg Tl(III) treatment were higher than for 12 mg/kg Tl(I) treatments. There were no significant differences for the TFs of stems and leaves among Ca treatments. The BF in the leaves of the blank Ca treatment were the lowest, being significantly lower than those from the 2.5, 3.0 g/kg Ca with 12 mg/kg Tl(I) treatments. Similarly, the BFs for the non-Ca treatment were also significantly lower than those from the 2.0, 2.5, 3.0 g/kg Ca with 8 mg/kg Tl(III) treatments. However, one interesting finding was that the BFs of the roots and stems for the blank Ca treatment were higher than those from the 1.0, 1.5, and 2.5 g/kg Ca with 12 mg/kg Tl(I) treatments and also higher than those from the 1.0, 1.5, and 3.0 g/kg Ca treatment with 8 mg/kg Tl(III) treatments. Furthermore, they were also higher than the average values. This suggests that the Ca treatment can significantly improve the ability of green cabbage to transfer Tl from the stems to the leaves.

A previous study reported that Tl(I) is easier to absorb by plant roots rather than Tl(III), because Tl(I) can replace K^+^ in the metabolism of plants and enter the plant tissue, and Tl(III) can only be taken through ion exchange and diffusion into the plant [60]. However, the results of the present study show that there was no significant difference in the BF values between 12 mg/kg Tl(I) and 8 mg/kg Tl(III) treatments (*p* < 0.05). Conversely, the BF values of leaves and whole plants in the 8 mg/kg Tl(III) treatment were higher than for the 12 mg/kg Tl(I) treatment (Table 5). This is inconsistent with the general position that Tl(I) mobility and bioavailability are higher than those of Tl(III) [49]. In addition to different concentrations of spiked Tl, the results may also be constrained by the different solubilities of TlCl (solubility is 2.9 g/L) and Tl(NO_3_)_3_ (solubility is 95.5 g/L) [61,62]. Furthermore, Tl(III) is thermodynamic unstable and could be reduced to Tl(I), then bio-absorbed by the root [63,64].

### 3.2. Mechanism of Ca-Induced Phytoextraction

The extractable concentrations of Ca increased with concentrations of Ca treatment, and increased by 57% in the 3.0 g/kg Ca treatment relative to the blank control. The solubility concentrations of Tl as a function of Ca-spiked treatment are shown in Figure 5.

The soluble concentrations of Tl increased with the concentrations of Ca treatment, and Tl solubility significantly elevated in the Ca treatment relative to deionized water. The soluble concentrations of Tl significantly increased by 210% and 150% in 3.0 g/kg Ca with 12 mg/kg Tl(I) and 8 mg/kg Tl(III) treatment, respectively, relative to the blank control. The soluble concentrations of Tl in Tl-polluted soils greatly increased by 7500% in 3.0 g/Ca treatment relative to the free Ca control. The results indicate that added Ca could replace Tl present on soil particles, resulting in higher Tl concentration in soil solutions. This finding is in agreement with [65], who found that if Ca is applied at high rates in Ca-rich fertilizers, it can replace Cd in soils, resulting in higher soluble Cd in the soil solution. At the same time, the ability of Ca to replace Tl is constrained by soil CEC. When the spiked Ca reaches 2.0 g/kg, both the extractable Ca in soils and the replaced Tl in soil solutions tend to be stable (Figure 2). This can be seen in the results of the total Tl mass of green cabbage stabilized in 2.0, 2.5, 3.0 g/kg Ca treatment experiments (Figure 4).

Tl(I) can replace K^+^ to enter the plant tissue, and then follows the uptake pathways of Na^+^-K^+^ ATPase and the K^+^-voltage gated channel, and activation of Ca^2+^ is more conductive to this process [66,67]. This may partly explain Ca-induced phytoextraction.

The geochemical speciation of Tl in soil before and after the pot experiments was reported in the previous study [63]. The percentage of water soluble, weak acid soluble, and reducible fractions to total Tl followed an ascending order in different treated soils, i.e., Tl + Ca treated soils > Tl treated soils > initial soils. Conversely, the percentage of oxidizable and residual fractions to total Tl followed a descending order in different treated soils, i.e., Tl + Ca treated soils < Tl treated soils < initial soils. However, the percentage of the labile Tl including water soluble, weak acid soluble, reducible, and oxidizable fractions to total Tl significantly increased in soils with the Tl + Ca-treated or Tl treated pots, relative to the initial soils (Figure 6).

Compared with Tl(III), Tl(I) is thermodynamically more stable, less reactive, and has higher mobility [68]. However, the data of the geochemical speciation of Tl in soils showed no significant difference in the percentage of labile Tl between 12 mg/kg Tl(I) and 8 mg/kg Tl(III) treatments, even in the water soluble and weak acid soluble fractions. Identically, there was also no significant difference in the percentage of labile Tl between 12 mg/kg Tl(I) and 8 mg/kg Tl(III) with 3 g/kg Ca treatments. The previous study showed that Tl in soil solutions mainly existed as Tl(I), and the distribution of Tl(I) and Tl(III) in soils is a dynamic process [63,69]. There was no significant difference among Tl(I) and Tl(III) treatments, which may be attributed to the fact that Tl(III) enters the soil solution after being reduced to Tl(I), and then taken up by the plant.

After planting, the pH values did not significantly differ among different Ca treatments for the initial soils (Figure 7). For the 12 mg/kg Tl(I) treated soils, after planting, the pH values were not significantly different among the 0, 1.0, and 1.5 g/kg Ca treatments (*p* < 0.05) but were significantly higher than those of the 2.0, 2.5 and 3.0 g/kg Ca treated soils. A similar result was observed in the 8 mg/kg Tl(III) treated soils. The pH values dropped with the increasing Ca concentration treatment in the 12 mg/kg Tl(I) and 8 mg/kg Tl(III) treatment soils. More acidic pH values in soil can reduce the negative charge on soil and increase the mobilization of Tl [70,71]. This may improve the efficiency of plant uptake of Tl from soil. Compared with the initial soils, the pH values of the experiment soils after planting were significantly higher in rhizosphere soils. These results can be explained by previous studies that reported higher pH (9.0) in rhizosphere soils caused by green cabbage root exudates [72].

In the pot experiments, we used Ca(NO_3_)_2_·4H_2_O as the Ca treatment solution. The concentration of N increases with the addition of Ca(NO_3_)_2_·4H_2_O to soil, but the higher contents of N negatively influence the yield of plants and the Tl uptake [72]. This further supports that Ca enhances thallium uptake in green cabbage.

## 4. Conclusions

The increasing Ca treatment resulted in a higher biomass of green cabbage in the pot trials (the biomass of whole plant increased by a maximum of 78.6%) and elevated the Tl contents in the leaves of green cabbage (the concentration of Tl in the leaves increased by a maximum of 105%). The migration of Tl from stems to leaves in green cabbage is strongly enhanced by the application of Ca to the soils. Ca spiked to soils significantly elevates the concentration of Tl in the water soluble and weak acid soluble fractions, which can be attributed to the geochemical process of the enhanced soluble Ca that may replace Tl bound to the soil particles, releasing Tl into the soil solution. The pH values dropped with the increasing Ca concentration treatment. Lower pH values can increase the mobilization of Tl in soil, which results in an increase of Tl taken up by the plants. Overall, the data presented in this work provide evidence that elevated Ca can enhance the uptake of Tl by green cabbage, which is of great significance for the improvement of agricultural safety management practices for the farming of crops in Tl-polluted areas and for the development of efficient technologies for the remediation of Tl-polluted sites.

## Figures and Tables

**Figure 1 ijerph-20-00004-f001:**
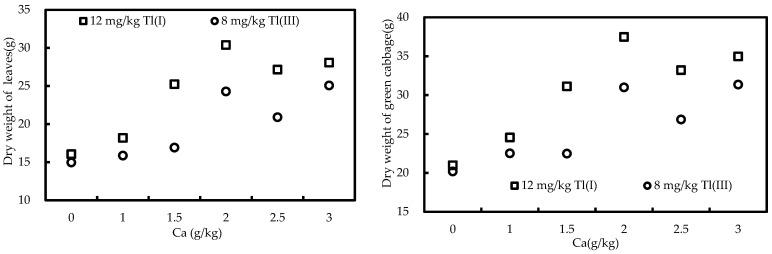
The relationship between the dry weight of plants (leaves, green cabbage) and the spiked Ca.

**Figure 2 ijerph-20-00004-f002:**
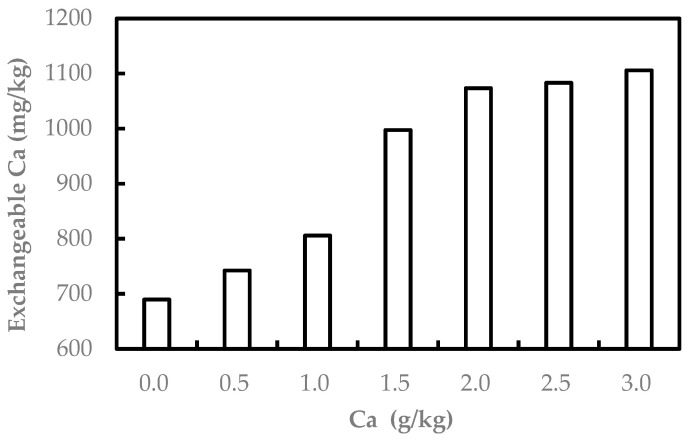
The exchangeable concentration of Ca in Ca-spiked pot soils.

**Figure 3 ijerph-20-00004-f003:**
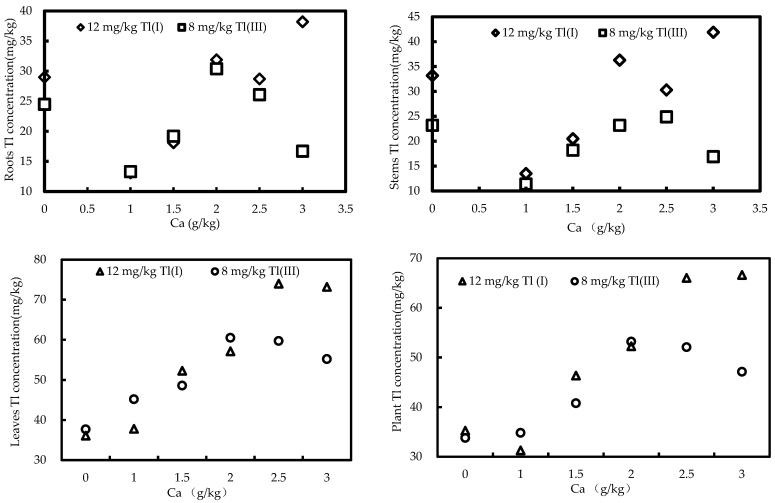
The relationship between the Tl in plants (roots, stems, leaves, whole plants) and the spiked Ca in pot soil.

**Figure 4 ijerph-20-00004-f004:**
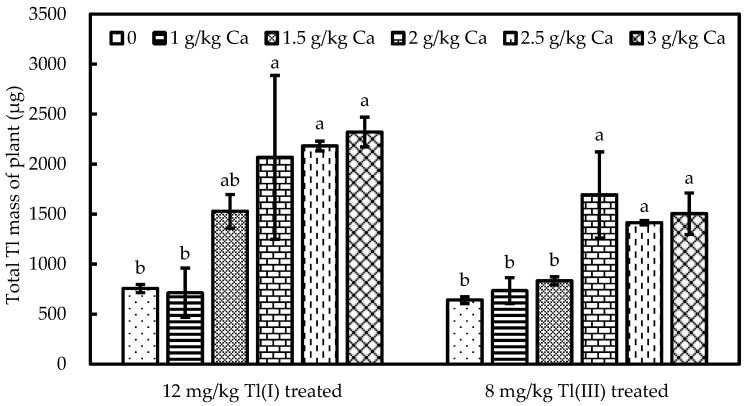
The exchangeable concentration of Ca in Ca-spiked pot soils. Bars denote standard deviation from means of three replicates. Significant differences among different Ca solutions are indicated by lowercase (*p* < 0.05).

**Figure 5 ijerph-20-00004-f005:**
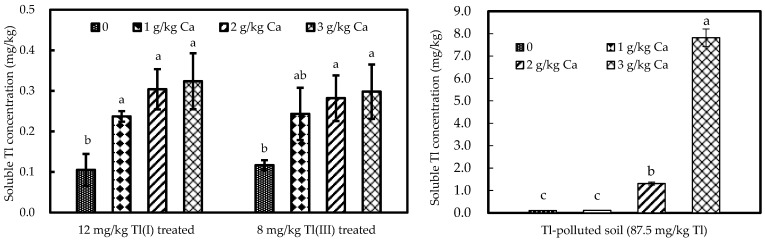
The concentration of Tl chemically solubilized by Ca. Bars denote standard deviation from means of three replicates. Significant differences among different Ca solutions are indicated by lowercase (*p* < 0.05).

**Figure 6 ijerph-20-00004-f006:**
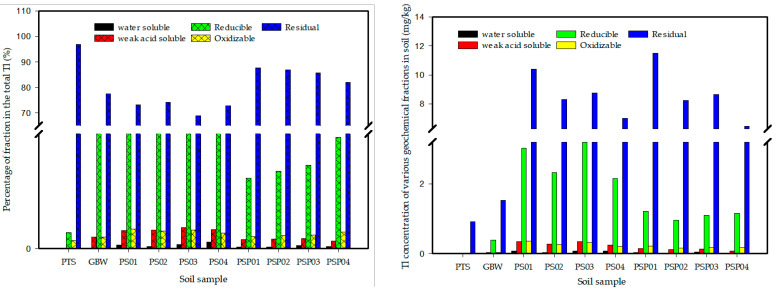
Tl existing in various fractions in soils, modified after [63]. PTS = Pot trial soils, GBW = GBW-07405, PS01 = 12 mg/kg Tl(I) treated, PS02 = 8 mg/kg Tl(III) treated, PS03 = 12 mg/kg Tl(I) + 3 g/kg Ca treated, PS04 = 8 mg/kg Tl(III) + 3 g/kg Ca treated, PSP01 = 12 mg/kg Tl(I) + Planted, PSP02 = 8 mg/kg Tl(III) + Planted, PSP03 = 12 mg/kg Tl(I) + 3 g/kg Ca + Planted, PSP04 = 8 mg/kg Tl(III) + 3 g/kg Ca + Planted.

**Figure 7 ijerph-20-00004-f007:**
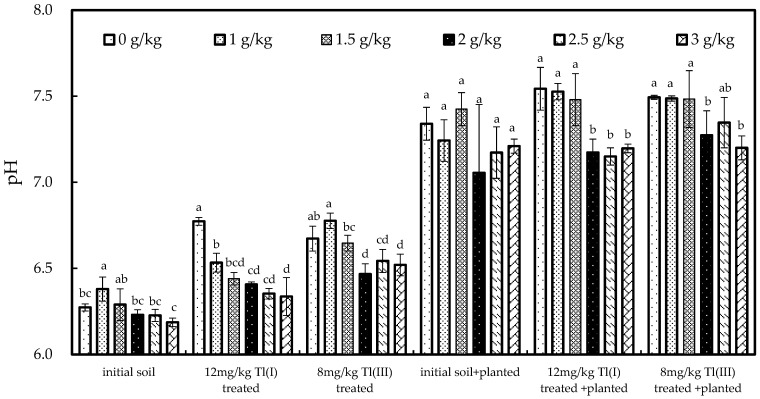
The pH values of soil before and after the experiment. Bars denote standard deviation from means of three replicates. Significant differences among different Ca treatments are indicated by lowercase (*p* < 0.05).

**Table 1 ijerph-20-00004-t001:** Physicochemical properties of the pot soil (mean ± sd, n = 3).

Soil Parameters	
pH (1:2.5)	6.27 ± 0.02
SOM (g/kg) ^1^	70.3 ± 1.48
CEC (cmol/kg) ^2^	21.5 ± 0.87
Particle size distribution	
sand % (>0.05 mm)	43.5 ± 0.77
silt % (0.002–0.05 mm)	47.4 ± 0.22
clay % (<0.002 mm)	9.05 ± 0.55
Total Ca (g/kg)	9.85 ± 0.64
Exchangeable Ca (mg/kg)	660 ± 43.1
Total Tl (mg/kg)	0.84 ± 0.05

^1^ SOM: soil organic matter. ^2^ CEC: cation exchange capacity.

**Table 2 ijerph-20-00004-t002:** Mineral composition of the initial pot soil.

Soil Mineral Type	Quartz	Smectite	Illite	Kaolinite	Feldspar	Calcite	Iron Mineral
Percentage composition (%)	87	4.7	1.5	0.96	1.7	3.0	1.2

**Table 3 ijerph-20-00004-t003:** Dry weight of the roots, stems, young leaves, old leaves, and whole plants of a single plant with 12 mg/kg Tl(I) or 8 mg/kg Tl(III) and various Ca-spike treatments (mean ± sd, n = 3, g).

Ca Treatment	12 mg/kg Tl(I)					8 mg/kg Tl(III)				
	Roots	Stems	Young Leaves	Old Leaves	Whole Plants	Roots	Stems	Young Leaves	Old Leaves	Whole Plants
control	1.3 ± 0.4 b	3.6 ± 0.3 b	8.7 ± 2.4 b	7.4 ± 1.2 c	21.0 ± 3.6 c	1.9 ± 0.1 a	3.4 ± 0.2 c	7.8 ± 1.2 c	7.2 ± 1.3 c	20.2 ± 2.6 c
1.0 g/kg	2.0 ± 1.1 ab	4.4 ± 0.1 ab	8.6 ± 1.6 b	9.6 ± 3.1 bc	24.6 ± 5.7 bc	1.5 ± 0.3 a	5.2 ± 1.2 a	8.0 ± 1.6 c	7.8 ± 1.7 bc	22.5 ± 3.6 bc
1.5 g/kg	1.9 ± 0.2 ab	4.0 ± 0.5 b	12.9 ± 4.4 ab	12.3 ± 0.4 ab	31.1 ± 5.1 ab	1.6 ± 0.6 a	4.0 ± 0.2 bc	9.2 ± 1.8 bc	7.7 ± 2.3 bc	22.5 ± 3.9 bc
2.0 g/kg	2.7 ± 0.7 a	4.4 ± 0.9 ab	14.6 ± 3.6 a	15.7 ± 3.8 a	37.5 ± 8.3 a	1.6 ± 0.6 a	5.1 ± 0.9 ab	12.2 ± 1.8 a	12.1 ± 0.4 a	31.0 ± 3.5 a
2.5 g/kg	2.2 ± 0.5 ab	3.9 ± 0.3 b	13.1 ± 2.2 ab	14.1 ± 1.8 a	33.2 ± 1.9 ab	1.5 ± 0.2 a	4.5 ± 0.1 abc	10.7 ± 1.4 ab	10.2 ± 1.3 ab	26.9 ± 0.6 ab
3.0 g/kg	1.9 ± 0.6 ab	5.0 ± 0.2 a	14.7 ± 0.9 a	13.4 ± 0.3 ab	35.0 ± 0.8 a	1.6 ± 0.4 a	4.7 ± 0.5 ab	12.5 ± 0.6 a	12.6 ± 0.2 a	31.3 ± 0.9 a

Significant differences among different Ca treatments are indicated by lowercase (*p* < 0.05).

**Table 4 ijerph-20-00004-t004:** Concentration of Tl in the roots, stems, young leaves, old leaves, and whole plants of a single plant with 12 mg/kg Tl(I) or 8 mg/kg Tl(III) and various Ca-spike treatments (mean ± sd, n = 3, mg/kg).

Ca Treatment	12 mg/kg Tl(I)				8 mg/kg Tl(III)			
	Roots	Stems	Leaves	Whole Plants	Roots	Stems	Leaves	Whole Plants
control	29.0 ± 4.4 ab	33.2 ± 2.8 b	36.1 ± 3.2 b	35.3 ± 3.1 bc	24.5 ± 3.3 ab	23.2 ± 6.9 a	37.7 ± 4.7 b	33.8 ± 4.6 b
1.0 g/kg	13.1 ± 3.7 bc	13.5 ± 0.0 c	37.8 ± 9.5 b	31.3 ± 7.0 c	13.3 ± 6.1 b	11.4 ± 0.7 b	45.2 ± 2.4 ab	34.8 ± 2.7 b
1.5 g/kg	18.1 ± 1.9 bc	20.5 ± 4.4 c	52.3 ± 0.9 ab	46.4 ± 1.5 abc	19.2 ± 3.4 ab	18.2 ±1.2 ab	48.6 ± 5.9 ab	40.8 ± 3.5 ab
2.0 g/kg	31.9 ± 6.7 a	36.3 ± 2.0 ab	57.1 ± 24 ab	52.3 ± 18 ab	30.4 ±7.6 a	23.2 ± 1.4 a	60.5 ± 14.6 a	53.2 ± 12 a
2.5 g/kg	28.7 ± 4.4 ab	30.3 ± 5.0 b	74.0 ± 0.4 a	66.0 ± 1.0 a	26.1 ± 3.7 ab	24.9 ± 5.1 a	59.7 ± 1.3 a	52.1 ± 0.03 a
3.0 g/kg	38.2 ± 8.3 a	41.9 ± 2.5 a	73.2 ± 5.4 a	66.6 ± 3.9 a	16.7 ± 8.3 ab	16.9 ± 7.2 ab	55.2 ± 5.5 a	47.1 ± 5.4 ab

Significant differences among different Ca treatments are indicated by lowercase (*p* < 0.05).

**Table 5 ijerph-20-00004-t005:** Transfer coefficient (TC) ^1^, bioconcentration factor (BF) ^2^ for green cabbage with 12 mg/kg Tl(I) or 8 mg/kg Tl(III) and various Ca-spike treatments (mean ± sd, n = 3).

Ca Treatment	12 mg/kg Tl(I) Treatment						8 mg/kg Tl(III) Treatment					
	TC_Stems_	TC_Leaves_	BF_Roots_	BF_Stems_	BF_Leaves_	BF_Whole plants_	TC_Stems_	TC_Leaves_	BF_Roots_	BF_Stems_	BF_Leaves_	BF_Whole plants_
control	1.2 ± 0.1 a	1.3 ± 0.1 a	2.3 ± 0.3 ab	2.6 ± 0.2 b	2.8 ± 0.2 b	2.8 ± 0.2 cd	0.9 ± 0.2 a	1.5 ± 0 a	2.8 ± 0.4 ab	2.6 ± 0.8 a	4.3 ± 0.5 b	3.8 ± 0.5 b
1 g/kg	1.1 ± 0.3 a	3.1 ± 1.6 a	1.0 ± 0.3 c	1.1 ± 0.0 c	3.0 ± 0.7 b	2.4 ± 0.5 d	1.0 ± 0.5 a	3.7 ± 1.5 a	1.5 ± 0.7 b	1.3 ± 0.1 b	5.1 ± 0.3 ab	4.0 ± 0.3 b
1.5 g/kg	1.1 ± 0.4 a	2.9 ± 0.4 a	1.4 ± 0.1 bc	1.6 ± 0.3 c	4.1 ± 0.1 ab	3.6 ± 0.1 bcd	1.0 ± 0.1 a	2.6 ± 0.8 a	2.2 ± 0.4 ab	2.1 ± 0.1 ab	5.5 ± 0.7 ab	4.6 ± 0.4 ab
2 g/kg	1.2 ± 0.2 a	1.9 ± 1.1 a	2.5 ± 0.5 a	2.8 ± 0.2 ab	4.5 ± 1.9 ab	4.1 ± 1.4 abc	0.8 ± 0.2 a	2.1 ± 1.0 a	3.5 ± 0.9 a	2.6 ± 0.2 a	6.9 ± 1.7 a	6.0 ± 1.3 a
2.5 g/kg	1.1 ± 0.0 a	2.6 ± 0.4 a	2.2 ± 0.3 ab	2.4 ± 0.4 b	5.8 ± 0.0 a	5.2 ± 0.1 ab	1.0 ± 0.1 a	2.3 ± 0.4 a	3.0 ±0.4 ab	2.8 ± 0.6 a	6.8 ± 0.1 a	5.9 ± 0.0 a
3 g/kg	1.1 ± 0.2 a	2.0 ± 0.6 a	3.0 ± 0.6 a	3.3 ± 0.2 a	5.7 ± 0.4 a	5.2 ± 0.3 a	1.0 ± 0.1 a	3.7 ± 1.5 a	1.9 ± 0.9 ab	1.9 ± 0.8 ab	6.3 ± 0.6 a	5.4 ± 0.6 ab
mean	1.1	2.3	2.1	2.3	4.3	3.9	0.9	2.7	2.5	2.2	5.8	5.0

^1^ TC = concentration of Tl in aboveground plants/concentration of Tl in roots of plants. ^2^ BF = concentration of Tl in each plant part/concentration of Tl in soils. Significant differences among different Ca concentration treatments are indicated by lowercase (*p* < 0.05).

## Data Availability

The associated dataset for the study is available upon request from the corresponding author.

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
