# Peer review of "Calcium Enhances Thallium Uptake in Green Cabbage (Brassica oleracea var. capitata L.)"

_ijerph, 2022, doi:10.3390/ijerph20010004_

Round 1

Reviewer 1 Report

The paper deals with the effect of calcium on the uptake of thalium in green cabbage. Authors stated that calcium in soils could significantly enhance thalium uptake and that the lower pH in soils also increases its mobility, which resulted in its accumulation in green cabbage.

The topic of paper is interesting. on the other hant, the manuscript should be improved before its acceptance and publication:

1) to explain better motivation for this research and relevance for practical application;

2) to describe BCR procedure in detail;

3) to discuss more the relevance of oxidation number on the behavior of thalium in soil (its mobility and uptake);

4) it is not clear why are experimental data in Fgs. 1 and 3 fitted by lines; the dependencies are not linear!

5) conclusions are too general, they should be concretized.

Reviewer 2 Report

This manuscript studied the influence of exogenous Ca in soils on Tl uptake in green cabbage. The experimental analysis is reliable, and the workload is abundant. However, the introduction and the conclusion should be improved. Furthermore, some tables and figures should be modified.

The following are the detailed opinions of the article:

Abstract:

1) Page 1, lines 11-12. This part does not describe the background and importance of this study.

2) Page 1, lines 12-13. “We investigated, for the first time, the influence of Ca in soils on Tl uptake in green cabbage and the associated geochemical process.” Please verify this, and I do not think it is an accurate description.

3) Page 1, line 20. “More interestingly, the BFs of the whole plant……” Please don’t use the abbreviation of BFs when it is used for the first time.

4) Page 1, lines 25-26. The conclusion of this study should be summarized. It’s just a general statement for similar studies.

Introduction:

5) The introduction needs to be sufficient. The risk of TI for both humans, plants, and the ecology should be listed there. Moreover, the author should consider why they conducted this study; more is needed in this manuscript.

Materials and Methods

6) Page 2, lines 68-69. Wrong sentence.

7) Why not use a three-line table?

Results

8) Figure 2. should be redrawn.

9) Figure 5. and Figure 7. need to be clarified.

10) Figure 6. Inconsistent fonts.

 Conclusions

11) Please enhance the discussions and condense the conclusions.

Reviewer 3 Report

Authors are suggested to revise the manuscript to reduce plagiarism and duplicate submission to below 15% and resubmit.

Round 2

Reviewer 1 Report

I agree with the publication of paper.